

# Assessing atoll island future habitability in the context of climate change using Bayesian networks

Mirna Badillo-Interiano [1, 2], Jérémy Rohmer [1], Gonéri Le Cozannet [1], Virginie Duvat [2, 3]

[1] BRGM, 45000, Orléans, France
[2] UMR LIENSs, La Rochelle University-CNRS, 7266, La Rochelle, France
[3] Institut Universitaire de France, France

*Correspondence to*: Mirna Badillo-Interiano (m.badillo@brgm.fr)

**Abstract.** Atoll islands are threatened by multiple climate change impacts, such as sea-level rise and extreme sea-level events, and ocean warming and acidification. A recent approach to assess climate change risk to these islands is to use multi-criteria expert judgment methods. These approaches can serve as a basis to the development of Bayesian Networks (BNs) integrating expert knowledge and uncertainties to perform climate risk assessments. Here, we use the model structure and expert knowledge of (Duvat et al., 2021), who assessed future risk to habitability for four Indian and Pacific Oceans' atoll islands, in order to discuss the advantages and limitations of the BN model. Advantages of the approach include the explicit treatment of uncertainties and the possibility to query expert knowledge in a non-trivial manner. For example, expert knowledge can be used to assess risks to habitability and future uncertainties and to explore inverse problems such as which drivers can exceed specific risk thresholds. Our work suggests that BN, though requiring a certain level of implementation expertise, could be used to assess climate change risk and support climate adaptation.

Keywords: climate change risk, atoll islands, Bayesian networks, uncertainties, climate adaptation

## 1. Introduction

Atoll islands are highly vulnerable to climate change. This vulnerability is mainly due to their low-lying elevation and limited land resources (Mycoo et al., 2022). The Intergovernmental Panel On Climate Change (IPCC) mentions in the Sixth Assessment Report that these islands are increasingly affected by multiple changes including sea level rise, increased temperatures, impacts of tropical cyclones, droughts, storm surges, coral bleaching, and changes in rainfall. Impacts related to these changes have already been observed, such as flooding, coastal erosion, and loss of coral reefs and ecosystem services. Multiple studies have shown that these impacts vary between island types and regions (Duvat et al., 2021) and can be exacerbated by compound effects (Wadey et al., 2017) and human activities (Rey et al., 2017).



Multiple projected changes, such as increased temperature, extreme sea levels, and degradation of ecosystems, are expected to exacerbate sea-related hazards in atoll settings (Mycoo et al., 2022). For example, Vitousek et al. (2017) suggest that changes in wave climate combined with sea-level rise will increase the risk of flooding, especially in the tropical Pacific and Indian Oceans. In some regions, increased drought intensity could result in freshwater insecurities (Karnauskas et al., 2016;

Schewe et al., 2014). Loss of coastal ecosystems and associated services resulting from the combination of global climate change and local anthropogenic disturbances will likely increase land loss, negatively impacting food and water supply as well as economic activities (Pratchett et al., 2008). This could challenge the ability of populations and ecosystems to recover and adapt in atoll settings.

In this context, integrated risk assessments become increasingly necessary to develop adaptation plans. However, such holistic assessments are complex mainly due to knowledge gaps, limited data, and uncertainties around climate change. Moreover, the complex interplay between climatic and non-climatic drivers turns on feedback loops and cumulative and cascading impacts (Simpson et al., 2021; Westra and Zscheischler, 2023) that are difficult to understand and predict. Recognizing this complexity, many previous integrated assessments assessed present-day or future climate risk following a

three-step approach. In a first step, a conceptual model identifying the different components of the studied system and their interlinkages was developed. Then, knowledge was collected using a multicriteria analysis to characterize the severity and confidence level of each driver of risk. In a final step, this knowledge was aggregated. In the area of coastal risk assessments, the "Coastal Vulnerability Index" of (Gornitz et al., 1991) is a foundational example of such approaches. More recent examples include the burning embers (Zommers et al., 2020) for climate risk assessments used in the Working Group 2

reports of the IPCC and the integrated system approach to assessing future climate risk to atoll islands developed by (Duvat et al., 2021).

Other approaches, such as Bayesian Networks (BNs), have shown potential to address complex systems and uncertainties. A BN is a probabilistic graphical model that allows for representing and quantifying interactions between multiple variables. In

coastal systems, BNs have been widely applied to understand physical processes such as coastal cliff erosion (Hapke and Plant, 2010), dune erosion due to extreme events (Heijer et al., 2012), and surf zone processes (Plant and Holland, 2011). These models have also been successfully used for hazard assessment and risk management. With this aim, BNs have been developed to predict multiple coastal hazards (Narayan et al., 2015; Poelhekke et al., 2016), to assess hurricane damages (van Verseveld et al., 2015), and to evaluate the effectiveness of risk reduction measures (Banan-Dallalian et al., 2023;

Ferreira et al., 2019; Jäger et al., 2018; Plomaritis et al., 2018; Sanuy et al., 2018). In small islands, there are few applications of BNs to assess hazards (Pearson et al., 2017) and the effectiveness of adaptation strategies (Sahin et al., 2021) (Table 1).

BNs explicitly integrate uncertainties, suggesting their potential to address climate change-related issues (Sperotto et al., 2017). Some applications include impact assessments of sea-level rise (Gutierrez et al., 2011; Yates and Le Cozannet, 2012)




and coral reef degradation (Baldock et al., 2019), and evaluations of adaptation strategies (Hafezi et al., 2020; Phan et al., 2020). Despite their growing application to climate change-related issues, applications remain limited, especially for small islands, and only a few focused on integrated climate risk assessments (Catenacci and Giupponi, 2013).

In this paper, we develop a BN model based on (Duvat et al., 2021) to assess the climate risk to the habitability of four atoll
islands in the Pacific and Indian Oceans (Figure 1) under the Representative Concentration Pathways (RCP) 2.6 and 8.5 for the years 2050 and 2090. We aim to explore the potential and limitations of the BN for integrated climate risk assessments. Our objective is to focus on the methodological aspects and not on collection of expert knowledge or the structuration of the problem. Therefore, we use the conceptual diagram of (Duvat et al., 2021) as a BN structure on the one hand, and their expert-based assessment to inform the conditional probabilities in the network on the other hand. We converted expert
judgments into probabilities (section 2.4.2) using beta distributions, which were then incorporated into the model as prior knowledge. The BN analysis (section 3) focuses on risk assessment, identification of major risk factors, identification of thresholds, and evaluation of the possible impact of adaptation measures. To study these elements, we addressed the following research questions:

• Risk assessment: What is the probability of risk to habitability given a RCP scenario and a time horizon? This question is addressed in section 4.1.
      • Identification of critical thresholds: What levels of risk could lead to adaptation limits? This question is addressed in section 4.2.
      • Identification of the major drivers of risk: Which risk factors are present when the risk to island habitability is high?
85       This question is addressed in section 4.3
      • Effectiveness of adaptation measures: To what extent is the risk to habitability reduced if we act on the risk factors that contribute the most? This question is addressed in section 4.4.

In the next sections, we present the framework and provide the fundamentals of BNs. This is followed by the description of
BN development and the results of the BN analysis. Finally, we discuss the sensitivity test results and the potential and limits of the method.





**Figure 1 Location of the atoll islands of interest: Male' and Nolhivaranfaru in the Indian Ocean, and Fogafale and Tabiteuea in the Pacific Ocean.**

**Table 1 List of reviewed studies of Bayesian network applications in small islands. Most of these studies integrate climate change to evaluate hazards and the effectiveness of adaptation strategies. However, to our knowledge, integrated risk assessments are not available or have not yet been conducted.**

| Type of application | Study | Model objectives | Conceptual model variables | |
|---|---|---|---|---|
| | | | Variables considered | Consideration of climate change |
| **Impact assessment** | (Baldock et al., 2019) | To assess the impact of reef degradation and climate changes on the shoreline | Hydrological, morphological | Sea-level rise, reef health |
| | (Uusitalo et al., 2012) | To assess the impact of environmental factors on coastal fish production | Ecological, environmental | Not considered |
| | (Callaghan | To predict wave propagation in coral reefs | Hydrological, | Sea-level rise |





| | | | | |
|---|---|---|---|---|
| | et al., 2018) | | morphological, climatic | |
| | (Pearson et al., 2017) | To estimate wave-induced flooding of reef-fronted coastlines | Hydrological, Morphological | Not considered |
| **Risk management** | (Hafezi et al., 2020) | To evaluate adaptation strategies for coral reef ecosystems | Anthropic, environmental, climatic | Rainfall pattern, sea surface temperature, sea-level rise, storm frequency, ocean warming, and acidification |
| | (Sahin et al., 2021) | To assess how ecosystem-based adaptations to climate change influence community wellbeing | Anthropic, environmental, socio-economic | Temperature, reef health |
| | (Sahin et al., 2019) | To predict coastal erosion and assess adaptation measures | Hydrological, geological, morphological | Sea-level rise |


## 2. Data and Methods

The proposed methodological framework is shown in Figure 2. Based on (Duvat et al., 2021), we present the BN model aimed at assessing the climate risk to the habitability of atoll islands and adaptation effectiveness under RCP 2.6 and RCP 8.5 in 2050 and 2090. The network structure is based on the conceptual model of atoll habitability from (Duvat et al., 2021).

The geographical settings are presented in subsection 2.2. The expert judgments and associated scores (our input data) are presented in subsection 2.3 (the full risk assessment database is available in the supplement provided by (Duvat et al., 2021)). Thus, the key methodological inputs in this manuscript are the translation of scores and confidence levels assessed by (Duvat et al., 2021) into probabilities on the one hand, and the development of the Bayesian network on the other hand.





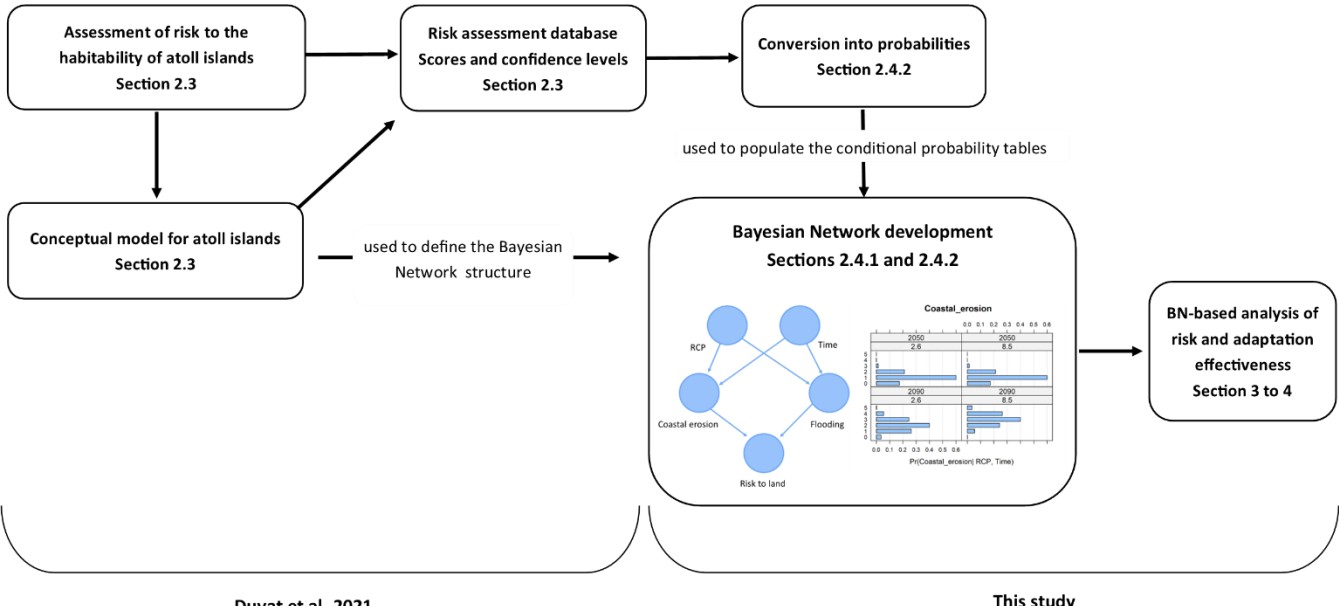

**Figure 2 Methodological framework.**

### 2.1. Bayesian networks

A Bayesian network (BN) is a probabilistic graphical model representing probabilistic associations between random variables. The BN is defined by a structure and conditional probability tables (CPT). The structure is defined by nodes and arcs, representing the variables and the relationships between these variables, respectively (Figure 3). The arrow indicates the influence direction from the *parent* node to the *child* node. Each node in the network is associated with a conditional probability table (CPT), which contains a local conditional distribution for that variable given its parents. In the case of no parents, the node is associated with a probability table and stores the prior probability.

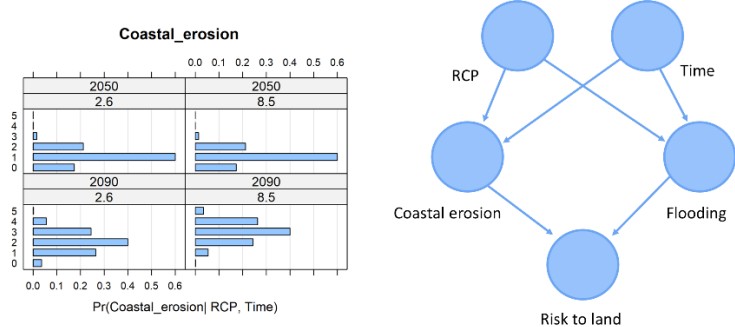

**Figure 3 Example of a BN structure composed of five nodes (right) and a CPT (left) containing the local conditional distribution of the variable Coastal erosion given its parents RCP and Time.**





We focus on discrete BN, in which the product of the local probability distributions of each node results in the joint
probability distribution function of all the variables $X = \{X_1, \dots X_n\}$ in the graph Eq. (1):

$$P(X_1, \dots X_n) = \prod_{i=1}^{n} p\big(x_i \big| pa(X_i)\big), \tag{1}$$

Where $pa(X_i)$ are the parents of $X_i$ (Pearl, 1988). The joint probability distribution function links all the variables in the BN,
therefore any change can be propagated through the network. This means that the BN can be used in a forward mode where
the changes are propagated from the child to parent nodes, but also in an inverse mode, from the parents to child nodes. This
flexibility is very useful to explore multiple combinations of events.

Bayesian inference relies on Bayes' theorem to compute posterior probabilities. According to Bayes' theorem, the probability
$p$ of an event R given the evidence O is given by Eq. (2):

$$p\big(R_i \big| O_j\big) = \frac{p(O_j | R_i)\, p(R_i)}{p(O_j)}, \tag{2}$$

The first term of the numerator is the likelihood, which is the probability of seeing the evidence given the event. The second
term is the prior probability, which is the probability of the event before the evidence. The denominator, known as the
normalization factor, is the marginal probability of the evidence.

### 2.2. Study islands

Duvat et al. (2021) and the present study focused on four contrasting atoll islands of the Pacific and Indian Oceans (Figure 1)
Male' (Maldives), Fogafale (Tuvalu), Nolhivaranfaru (Maldives), and Tabiteuea (Kiribati). These islands were chosen to
cover a contrasted range of geographical settings (Table 2). For example, they include the highly urbanized and densely
populated island of Male' in the Maldives and the rural and mostly non-armored island of Tabiteuea in Kiribati.

The main climate change drivers affecting these islands include changes in atmospheric temperatures and rainfall patterns,
sea-level rise, increasing sea surface temperature, coral bleaching and ocean acidification. Other significant drivers are
increased distant-source wave height, increased intensity of the most intense tropical cyclones and El Niño/La Niña events,
and increased intensity and frequency of marine heat waves (Duvat et al., 2021).



**Table 2 Description of the atoll islands of interest.**

| Atoll Island | Archetype | Economic activities | Implications for hazards | Sources |
|---|---|---|---|---|
| Male', North Kaafu Atoll, Maldives, Central Indian Ocean | Urban, densely developed, and entirely protected by engineered structures | Tourism, fisheries, and agriculture (~2% in 2014) | Low coastal erosion susceptibility due to the engineered structures. Flooding can occur due to extreme wave conditions and high sea levels. The risk of deterioration of coastal protection, partial loss of some areas, and flooding may increase this century due to the combination of sea level rise with increased wave height. | (Duvat et al., 2021) (Wadey et al., 2017) |
| Fogafale, Funafuti Atoll, Tuvalu, Western Pacific Ocean | Urban, highly modified by human activities. Limited coastal protection | Fisheries | Coastal erosion in some shoreline sections. High flooding susceptibility due to the low-lying elevation, local human disturbances, and inefficient protection structures. The most developed and populated zones are in vulnerable areas. High risk of coastal erosion, flooding, and reduction in reef fish production are projected through to 2090. | (Duvat et al., 2021) (Yamano et al., 2007) |
| Tabiteuea, North Tarawa, Kiribati, Western Pacific Ocean | Rural, mostly natural, with nearly entirely natural sand shorelines. Limited coastal protection | Fisheries and agriculture | Susceptibility of flooding risk due to the potential combination of state of POD/ENSO/ annual tidal extremes/spring tides and meteo-ocean events. Populated zones are in vulnerable areas. Moderate to high coastal erosion and flooding are expected through 2090. Other potential risks include a reduction in reef fish production and biomass of tuna. | (Duvat et al., 2021) |
| Nolhivaranfaru, Maldives | Rural, mostly natural, with extensive vegetation cover | Fisheries and agriculture | The island is relatively stable in land area. High susceptibility to flooding due to its low-lying elevation (1m). Moderate to high flooding is expected through 2090. | (Duvat et al., 2021) |

## 2.3. Expert judgment and assessment database

Duvat et al. (2021) assessed the climate related risk to future atoll habitability under two RCP scenarios. The assessment relied on a comprehensive literature review, available dataset analysis, and expert judgment. First, based on peer-reviewed



scientific papers and recent IPCC reports the authors identified five major Habitability Pillars (HPs): availability of sufficient land; supply of safe freshwater; supply of nutritious food from local and/or imported sources; access to safe settlements and infrastructure; and access to sustainable economic activities (Figure 4). Then, they defined a set of Risk Criteria (RC) considered as the major factors contributing to the risk to Habitability Pillars in study islands. For example, *coastal erosion* and *flooding* were identified as the main factors contributing to the risk to land.

After that, (Duvat et al., 2021) scored from 1 to 5 each risk criteria based on the expected severity of additional climate risk under both RCP and time horizons. For example, if the risk of *coastal erosion* was expected to be low, the risk level was scored as 1. Conversely, if a low-lying island was expected to experience a high risk of *flooding*, the risk level was scored as 5. The authors provide a detailed rationale for each score in their supplement (Duvat et al., 2021). The overall risk to habitability was calculated by aggregating the scores of risk criteria and habitability pillars. Six risk levels were considered:

undetectable (corresponding to no additional climate risk in the future compared to today's risk level), very low, low, moderate, high, and very high (corresponding to very high additional risk in the future compared to today's risk level). Moreover, each risk level was associated with a confidence level (very low to very high) based on evidence and the level of agreement (see Duvat et al., 2021 for details on the method).

       In this work, we use the risk assessment database, including the scores and confidence levels, as input data for the BN

model. Data pre-processing is detailed in the following sections.





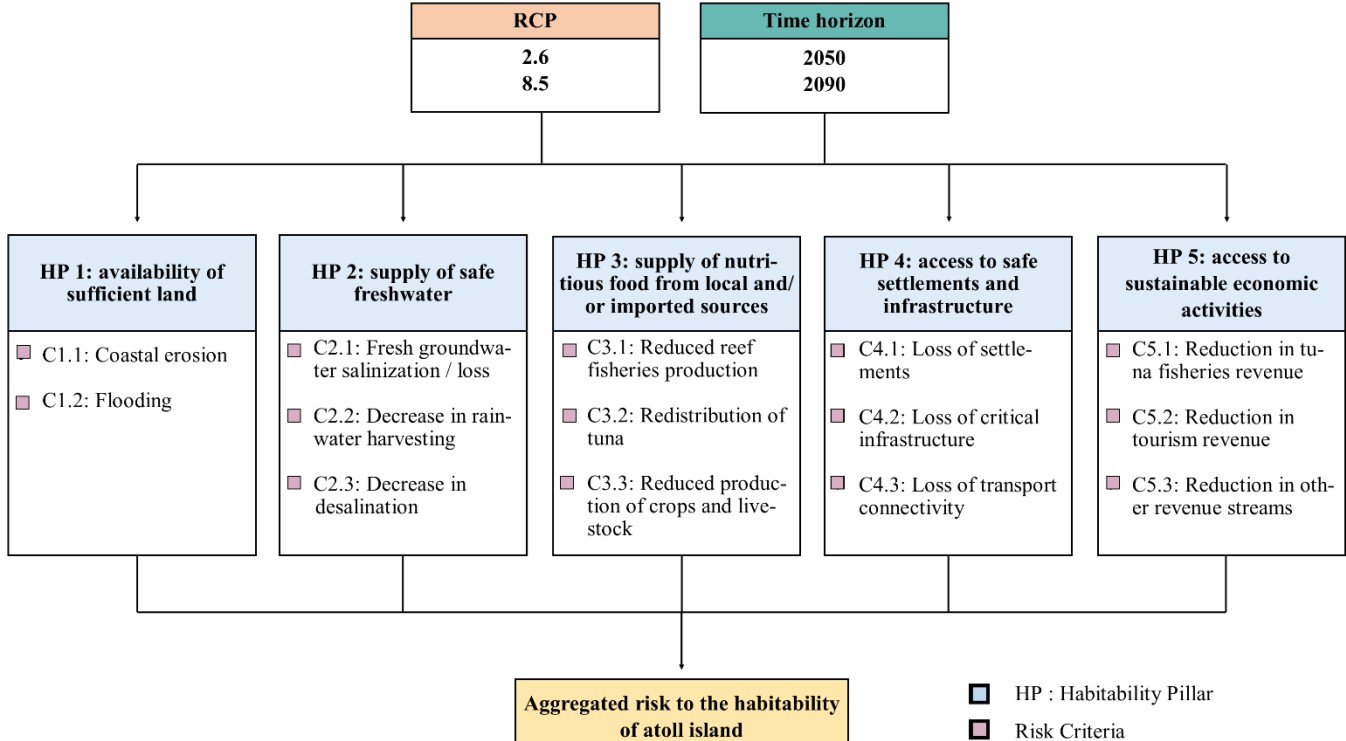

**Figure 4 Variables assessed by Duvat et al. (2021). The authors evaluated each Risk criterion based on the estimation of additional climate risk compared to today's risk level for the two RCP scenarios and time horizons. The risk for each habitability pillar results from the aggregation of risk criteria scores. The overall risk to island habitability results from the aggregation of the scores obtained for the five habitability pillars.**

### 2.4. BN development

#### 2.4.1.  Network structure

The network structure (Figure 5) is based on the conceptual model of atoll island habitability from (Duvat et al., 2021). It is a simplified representation of the relationships between the main factors contributing to risk to habitability in atolls. It comprises two parent nodes representing the RCP scenarios and the time horizons, influencing the thirteen risk criteria (RC). The RCs are linked to their corresponding Habitability Pillar (HP) and the five pillars are associated with the risk to island habitability. The BN structure is the same for all islands.





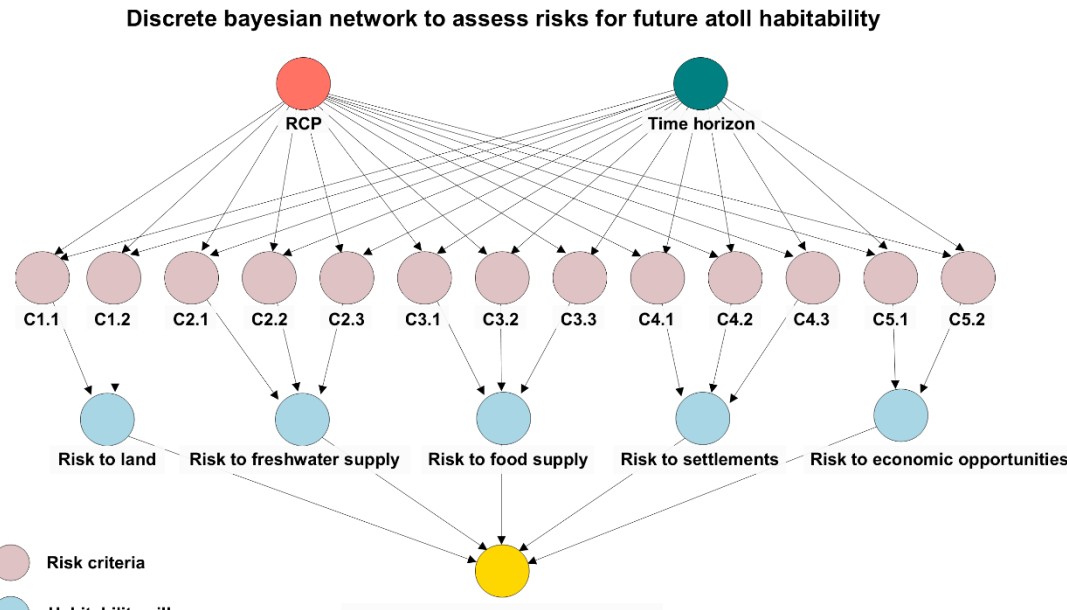

**Figure 5 Bayesian network developed for Male' atoll Island.**

**Table 3 Variables in the BN model and the discretization applied to each node. The states of the nodes correspond to the aggregation of risk levels.**

| Node | States |
|---|---|
| RCP | RCP 2.6 and RCP 8.5 |
| Time horizon | 2050 and 2090 |
| Risk criteria | 0 to 5 |
| Habitability Pillar | HP1 : 0 to 15 |
| | HP2: 0 to 15 |
| | HP3: 0 to 20 |
| | HP4: 0 to 20 |
| | HP5: 0 to 10 |
| Risk to Atoll Island Habitability | 0 to 100 |

### 2.4.2. Conditional probability tables

We populated the conditional probability tables with the risk levels and associated confidence levels provided by the experts. To do this, we translated the expert judgments into conditional probabilities using Beta distributions. We selected the Beta distribution (Ferrari and Cribari-Neto, 2004) due to its flexibility, as their possible shapes cover a range of possibilities that





span all potential cases that we considered in this study (Figure 6). Specifically, we anticipate the mode of the distribution to correspond to the primary score provided by (Duvat et al., 2021) with other scores associated with probabilities that decrease as they diverge from the main score. Additionally, as the confidence in the assessment decreases, we expect the probability distribution to become flatter. The Beta distribution effectively captures these characteristics, as illustrated in Figure 6. To represent the maximum confidence level (5) we used a Dirac distribution to put all the probability on the assessed risk level.


The selection of a particular distribution involves some subjectivity. To evaluate the impact of this subjectivity on our results, we supplemented our primary set of distributions with an alternative set that provides a more conservative interpretation of the weights. This secondary set was used to conduct a sensitivity analysis, the results of which are discussed in the discussion section. More details about the conversion of the scores and confidence levels into probabilities are

provided in the supplement (S1.2).



**Figure 6 Cumulative Beta distributions generated for each combination of risk and confidence levels. These distributions reflect the uncertainty in the expert risk assessment. A wider spread in the distribution indicates a lower confidence level. We used these**

**distributions to populate the conditional probability tables. The support of the distribution was divided into 6 intervals, one for each risk level. The dotted lines indicate the probability corresponding to each risk level.**

We populated the conditional probabilities tables associated with the *Habitability Pillars* and the *Risk to the Habitability* nodes in a deterministic way. We associated a probability of one with the aggregated risk level and a probability of zero with





the other levels. This approach allows us to remain consistent with the expert assessment and reduces computation time

when making inferences.

### 2.4.3. BN model validation

We validated the BN model using expert knowledge and the data from (Duvat et al., 2021) assessment database. We verified

that the model structure, discretization of variables, and parametrization were consistent with expert knowledge and existing literature on atoll socio-ecosystems. To validate the model behaviour, we verified that the outcomes of both the sub-networks and the entire network correctly reflected the risks assessed by experts. The validation process could be improved by integrating quantitative data such as measurements or numerical modeling results.

### 3. BN-based analysis

One of the main advantages of BNs is the possibility to interrogate the model in multiple ways and directions. The questions asked to the BN model are called probabilistic queries. These queries allow us to investigate multiple "what if" scenarios. To do this, we assume prior information and compute the probabilities of the variable of interest. Queries can be made in a direct way, this is from the parent to the child, or inverse mode, from the child node to the parent node. In this work, we focused on risk assessment, evaluation of risk criteria contribution, adaptation effectiveness, and identification of thresholds.

We translated our research questions into probabilistic queries as shown in Table 4.

**Table 4 Research questions and their translation into probability queries.**

| Research question | Probabilistic query | Section |
|---|---|---|
| What is the probability of risk to habitability given a RCP scenario and a time horizon? | P(Risk to island habitability \| RCP & Time horizon) | 4.1 |
| What levels of risk could lead to adaptation limits? | P(Risk to island habitability \| RCP & Time horizon & Risk criteria level) | 4.2 |
| Which risk factors are present when the risk to island habitability is high or very high? | P(Risk criteria \| RCP & Time horizon & Risk to island habitability = High or Very high) | 4.3 |
| To what extent is the risk to habitability reduced if we act on the risk factors that contribute the most? | In this experiment, we suppose the implementation of adaptation measures that reduce the risk factors to moderate levels. For example, flooding <= 2: P(Risk to island habitability\| RCP & Time horizon & Flooding = c(0,1,2)) | 4.4 |





We used the *cpquery* function from the bnlearn package (Scutari, 2010) to perform the queries. The inference algorithm used
was the *likelihood weighting* method. We chose this method because it better handles low probabilities.

## 4.    Results

### 4.1. Risk assessment and validation

In this experiment, we interrogated the BN model about the probability of risk to island habitability given an RCP scenario
and a time horizon. Figure 7 shows the probability of risk to habitability under the RCP 2.6 and 8.5 scenarios in 2050 and
2090 for the four islands. In 2050, the risks to habitability are similar in Male', Tabiteuea, and Nolhivaranfaru. The
probability distribution is slightly shifted toward higher levels for RCP 8.5 compared to RCP 2.6, the risk is low to moderate
in both scenarios. On the other hand, risks to island habitability are higher for Fogafale compared to other islands, the most
likely outcome being moderate risk levels. The contrast between RCP 2.6 and 8.5 is also more marked than in Male' and
Tabiteuea. In 2090, the risk to habitability shows more contrast between the two RCP scenarios. In the RCP 2.6 scenario,
Male', Tabiteuea, and Nolhivaranfaru are expected to experience low to moderate risk, and Fogafale is likely to experience
moderate to high risk. Under the RCP 8.5 scenario, all islands may experience moderate to high-risk levels.

As expected, these results are consistent with the aggregated risk to island habitability from (Duvat et al., 2021) (yellow
line), though small differences can be observed. These differences are due to the fact that the aggregated risks in (Duvat et
al., 2021) result from the combination of their best estimates and do not incorporate confidence levels. Conversely, our
approach interprets confidence levels as probability distributions around these best estimates. As shown in Figure 6, these
distributions can be skewed, resulting in slight shifts in the aggregated risk levels. Figure 7 shows that these differences are
small, and arguably smaller than the uncertainties resulting from the expert judgment itself. The results include a density
curve to smooth the peaks in the histograms. These peaks may be due to deterministic relationships. We populated specific
conditional probability tables in a deterministic way, associating a probability of one with the aggregated risk level and a
probability of zero with the other levels. The combinations matching the aggregated risk will have a higher probability
resulting in peaks at the distribution.







Figure 7 Posterior probability distributions for the RCP 2.6 and 8.5 scenarios in 2050 and 2090 for each island. The yellow line represents the aggregated risk to habitability assessed by (Duvat et al., 2021). The black curve is a smoothed version of the histograms.

## 4.2. Identifying critical thresholds

This experiment aims to identify which risk levels could lead to adaptation limits in some islands. Adaptation limits are defined as "the point at which an actor's objectives (or systems needs) cannot be secured from intolerable risks through





adaptive actions" (IPCC, 2023). These limits are related to the purple zone on burning embers diagrams, which show the changes in risk to humans and ecosystems as a function of global mean temperature (Zommers et al., 2020). The purple zone in these diagrams indicates a very high risk that can cause irreversible impacts and exceedance of adaptation limits. The burning embers diagrams have four risk categories: undetectable, low, moderate, and very high. In this work, we used the six risk categories defined by (Duvat et al., 2021) in which very high risk to habitability is > 80.

We explored the possibility for islands to reach adaptation limits thresholds. To do this, we interrogated the model about the probability of the risk to habitability under different risk criteria levels. Figure 8 illustrates the outcomes for Male' and Fogafale under RCP 8.5 in 2090. In Male', the results suggest that no risk criteria level could lead to exceeding the adaptation limits threshold (purple line). Conversely, in Fogafale severe risk criteria (levels 4 or 5) may lead to exceeding this threshold. Even at low risk criteria, the risk to habitability remains high. On this island, implementing adaptation measures that focus only on a specific risk reduction may not be sufficient.

This analysis also allows for identifying the risk criteria with the major contribution to the risk to habitability. This is reflected by the magnitude of the distribution shift. In Male' and Fogafale, variations on the risk level of *loss of settlements* generate a slight distribution shift and therefore a slight impact on habitability. In contrast, increments in *flooding* risk level have a more important contribution. These outcomes could be useful to identify major drivers and target possible adaptation strategies. For example, for these islands, *flooding* appears as the factor with a major contribution to the risk to habitability. In these cases, flooding reduction measures could be privileged.

The results for the other atoll islands are presented in the supplement (S2.1).





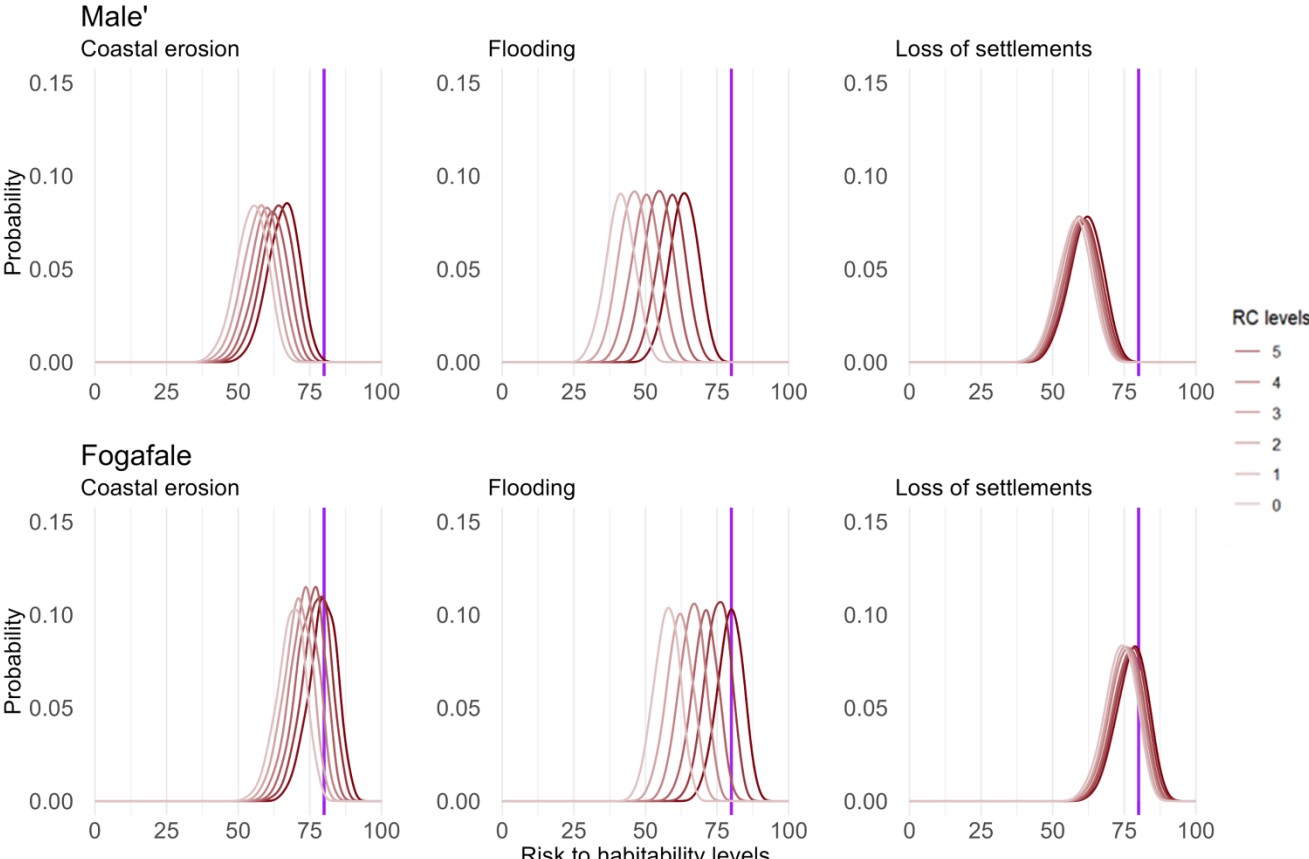

**Figure 8 Risk to Male' and Fogafale habitability under the RCP 8.5 in 2090. Each distribution represents the impact of different**
**risk criteria levels. For example, for a flooding risk of 1, the query is written as P(Risk to habitability | RCP = 8.5 & Time = 2090**
**& Flooding = 1). In Male', the adaptation limit (purple line) may not be exceeded under this scenario. In contrast, in Fogafale**
**severe risk criteria levels could lead to reaching adaptation limits.**

### 4.3. Identification of major drivers of risks

As mentioned in section 4.2, a very high risk to habitability could be reached at specific risk criteria levels. In this section,
we use the BN to perform inverse analysis. We explore the conditions that lead to very high risk to habitability, by
calculating the probability of each risk criteria level when the risk to habitability is very high.

Figure 9 shows the results for Fogafale under RCP 8.5 scenario in 2090. The probability distributions with and without the
constraint of very high risk to habitability are represented by red and gray bars respectively. Under a very high risk to
habitability, the probability distributions shift towards levels 4 and 5, indicating a correlation with severe risk criteria. This is



the case for multiple variables including *decrease in rainwater harvesting* and *desalination*, *flooding*, *reduced fisheries production,* and *loss of settlements and transport connectivity*. The variations between the distributions with and without the habitability constraint reflect the impact of the risk to habitability node on the risk criteria nodes and vice-versa. For example, when we constrain the risk to habitability to very high levels, there is a significant shift in the probability distribution of *flooding*. This reflects that a variation in habitability will significantly impact *flooding* and inversely, a variation in *flooding* will impact the risk to habitability.

In all atoll islands, we observe the same correlation between high risk to habitability with severe risk criteria. These results are presented in the supplement (S2.2).

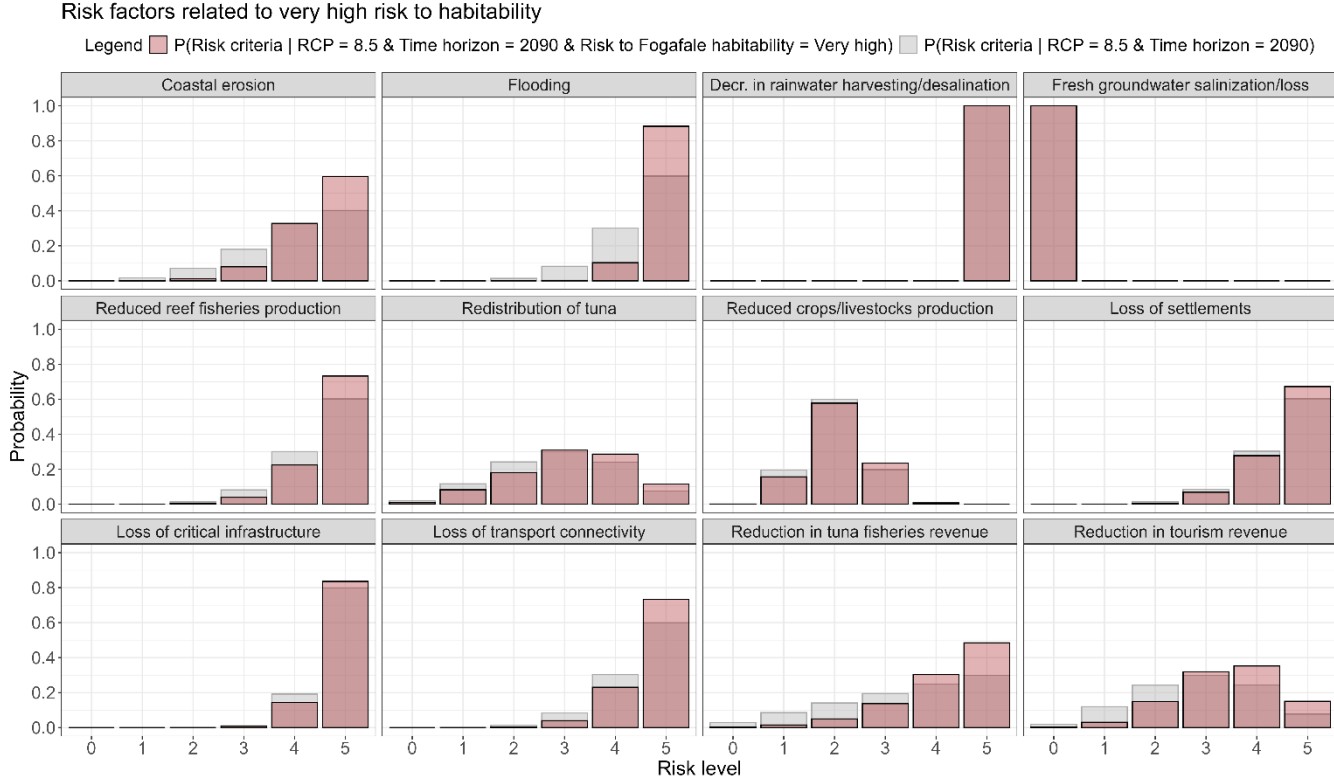

**Figure 9 Probability of risk criteria levels under the RCP 8.5 in 2090. The red bars represent the results when the risk to Fogafale habitability is very high (query: P(Risk criteria | RCP = 8.5 & Time horizon = 2090 & Risk to Fogafale habitability = Very high)). The grey bars represent the results without the habitability constraint (query: P(Risk criteria | RCP = 8.5 & Time horizon = 2090)). The results reflect which risk criteria are associated with high risk to Fogafale habitability. These include flooding, loss of critical infrastructure, reduced reef fisheries production and decrease in rainwater harvesting.**

### 4.4. Effectiveness of adaptation measures

In this experiment, the objective is to assess how the risk to habitability decreases when the key risk factors are reduced, especially those previously identified as major contributors to risk to habitability. To do this, we assume implementing





adaptation measures that reduce the risk level, such as protecting coastal development, adapting settlements, or managing

retreat. The impact of adaptation measures can be evaluated by calculating the probability of risk to habitability given a risk criteria level <= 2 under the RCP = 8.5 in 2090. In both islands, the results show that reducing *flooding* has a major impact on the risk to habitability (Figure 10). In Male', implementing an adaptation measure that reduces *flooding* to moderate levels could reduce the risk to habitability from high to moderate. These results suggest that an adaptation policy focused on reducing flood risk only can already provide substantial benefits to preserve the habitability of Male' Island. Yet, adaptation

focused on flood risk only would not be sufficient in all contexts. For example, Figure 10 shows that a panel of measures is necessary for Fogafale to keep the risk to habitability to a moderate level.

The results for the other atoll islands are presented in the supplement (S2.3).





Figure 10 Probability distributions of the risk to habitability given different adaptation measures under RCP 8.5 in 2090 in Male' and Fogafale. No adaptation measures (No-AM) are considered in the baseline scenario. The adaptation measures considered in this analysis are focused on the reduction of flooding (FL), coastal erosion (CE), and loss of settlements (LS). In both atoll islands, adaptation measures focused on flooding could significantly reduce the risk to habitability. In Male', reducing flooding or coastal erosion could reduce the risk to habitability from high to moderate level. However, in Fogafale, multiple risk reduction measures are needed to reach moderate risk to habitability.



## 5. Discussion

The BN model was developed based on the conceptual model and expert risk assessment provided by (Duvat et al., 2021).
The model allowed for the integration of qualitative information and uncertainties associated with expert judgments (see S1.2 in the supplement). Expert judgments included various environmental, economic, physical, and climatic variables that we incorporated (explicitly or implicitly) into our BN model, allowing for integrated analysis. The BN allowed us to analyse multiple problems, including risk assessment, identification of major risk factors, evaluation of risk reduction measures, and identification of thresholds.


### 5.1. Risk to habitability assessment using the BN

BN allows to derive a best estimate of the risk, but also to quantify the confidence by providing the 17$th$ and 83$rd$ percentile results (Figure S9 in the supplement) interpreted as likely ranges (to refer to a probability of at least 66%) according to the IPCC likelihood scale. By 2050, under RCP 2.6 and RCP 8.5 scenarios, the risk to habitability for Male', Tabiteuea, and
Nolhivaranfaru is likely to be low (Figure 7). Conversely, in Fogafale, higher risk is more likely. This is due to its very low elevation, limited coastal protection, and high exposure of habitats and critical infrastructures (Duvat et al., 2021). By 2090, the contrast between the RCP scenarios becomes more important. This is attributed to the divergence of climate projections during the second half of the 21st century, as well as the cumulative and cascading impacts of climate change (Duvat et al., 2021). Under the RCP 2.6 in 2090, the risk to habitability is likely to be low-to-moderate across all islands except Fogafale.
In contrast, under the RCP 8.5 the risk to habitability is likely to increase from moderate to high or very high levels. In Male', Duvat et al. (2021) attributed this increase to the expected increase of flooding and degradation of coastal protection. In rural islands, the expected high risk is associated with their high dependence on coastal ecosystem services for food (e.g., reef fish abundance), water supply (e.g., reduction of saltwater intrusion), land stabilization (e.g., sediment supply and wave impact reduction), and economic activities (e.g., tourism, fisheries, agriculture). Under the RCP 8.5, ecosystems are expected
to decline due to the exceedance of critical thresholds, including the temperature threshold for tropical reef-building corals (Cooley et al., 2022), as well as regional bleaching thresholds. According to Duvat et al. (2021), risk to habitability is expected to be higher in rural islands due to their limited capacity to manage climate-related impacts through the implementation of coastal adaptation, technology, and imports, compared to urban islands.

### 5.2. Complex queries within the BN

The BN method has the advantage of enabling users to make complex requests. Typically, these complex requests involve queries that assume specific priors and compute the probabilities using the BN in reverse mode. In other words, the use of BN here simplifies the investigation of "what if" scenarios. The identification of major drivers of risk presented in section



4.3 is an example of such inverse analysis, which could be useful to target high risk factors and thereby identify potential adaptation measures. Figure 9 shows that severe *flooding* and *loss of settlements* are highly correlated to a very high risk to habitability. We used this information to target the adaptation measures proposed in the experiment 4.4. This analysis also highlights that a high risk to habitability can arise from the interaction of multiple factors. Therefore, integrated risk assessments and multi-risk adaptation strategies are needed.

Finally, the risk reduction measures analysis highlights the potential of BNs as decision-support tools, as suggested by multiple authors (Ferreira et al., 2019; Jäger et al., 2018; Rachid et al., 2021). Duvat et al. (2021) evaluated climate risk considering the current level of adaptation, classified as moderate. Examples of currently implemented measures are food imports and water desalination to counter local resources decline, and hard protection to reduce flooding and coastal erosion. In our BN analysis, we assessed the potential impact of additional risk reduction measures. However, we did not assess their feasibility. Therefore, rather than suggesting a specific type, we assumed the implementation of adaptation measures adapted to the specific island context. Some examples of practices applied in small islands are accommodation, vertical adaptation, ecosystem-based measures, and hard protections (Mycoo et al., 2022). In this work, we only consider some examples of possible adaptation measures. However, we did not assess the practicalities of their local implementation. In future work, a wider range of local parameters including the elevation of the atoll island, and exposure to extreme events could be considered in the context of a wider framework involving social perception, economic feasibility, co-benefits, and trade-offs.

### 5.3. Limitations

The presented method has limitations. The first one was the translation of expert judgments into probabilities. To do this, we selected specific beta distributions that best reflect the expert knowledge (best estimate and confidence level). We conducted several tests to select beta distributions dispersed enough to represent the expert knowledge without overfitting the model. However, the selection of a specific distribution introduces a degree of subjectivity (in addition to that related to the nature of the data). To evaluate the impact of our choices, we carried out a sensitivity test using an alternative set of beta distributions (detailed in the supplement). For this alternative set, we associated a higher weight to the confidence level. This is reflected by a higher probability associated with the risk level assessed by experts. For example, an assessed risk level associated with low confidence was represented by a probability of 30% in the initial set, and a probability of 40% in the alternative set. The sensitivity test shows slight differences between the two sets. The BN analysis using the alternative set of beta distributions results in less dispersed distributions and slightly different median values. However, we obtained the same risk categories in both cases, suggesting that the results are relatively robust against the probabilistic interpretation of expert judgments.





Other limitations are related to 1) the BN structure, 2) variables discretization, and 3) inference methods. Regarding 1), we used a structure similar to that of (Duvat et al., 2021), where the scores are essentially added to compute an aggregated risk level. (Duvat et al., 2021) did not consider interactions between habitability pillars, and therefore we did not include these interactions in the structure of our BN. The habitability pillar *Risk to land* has an impact on *local freshwater* and *food land-based supply*, *settlements and infrastructure*, and *economic activities*. Not considering these interactions could result in an

underestimation of the risk to habitability. In future work, we will consider these relationships, but it must be considered that since BNs are based on direct acyclic graphs, their structure cannot explicitly account for positive or negative feedback. This can be a limitation in our approach if risk or adaptation problems with large positive or negative feedback are considered. Concerning 2), the discretization of continuous variables could be another source of uncertainty. Depending on the states of the variables, the discretization could lead to a loss of information (Rohmer, 2020). For example, the use of too large or too

short categories may overlook subtle variations in the system, reducing the accuracy and relevance of the results. The use of hybrid BNs including continuous and discrete variables (Beuzen et al., 2018) could be helpful to minimize these uncertainties. Regarding 3), in our BN analysis, we used the likelihood weighting method for inference. This method has some limitations including low precision for estimating low-probability scenarios (Scutari, 2010), which were not in the scope of this study. To improve computational efficiency, alternative sampling methods will be considered in future research

work (Yuan and Druzdzel, 2006).

## 6.   Conclusions

The objective of this work is to present a reproducible methodological framework to develop a Bayesian network model based on expert judgments. We used the model structure and expert knowledge of (Duvat et al., 2021), who previously assessed the risk to habitability for 2050 and 2100 under two contrasting RCP scenarios for four atoll islands in the Indian

and Pacific Oceans. We performed the same risk assessment using a BN model. Our objectives were to integrate uncertainties in the risk assessment and analyse the potential and limitations of this approach. The BN reflects the expert knowledge consistently and takes into account the associated uncertainty. The model allowed us to analyze "what-if" scenarios that could be useful to assess the impact of climate change and to identify potential risk reduction measures.

Bayesian networks are usually developed using data to define their structure and conditional probability tables. In cases when limited data is available, these models can be fully parameterized using expert knowledge. We demonstrate that when expert risk assessments are available, they can serve as a basis for BNs. Our work gathers the detailed risk assessment and meaningful relationships between variables provided by the experts with the uncertainties integration and analysis of multiple scenarios offered by BNs. Further work will address the limitations of this approach, including the BN structure,

variables discretization and inference methods. However, this first attempt highlights the potential of BNs as a



complementary tool for integrated risk evaluation in small islands and potentially in other adaptation problems involving complex socio-ecosystems and expert judgment.

**Code and data availability**

The data and code used in this study are available at https://github.com/MirnaBadillo/Bayesian_Network_Risk_to_Atoll_Island_Habitability.

**Author contribution**

Conceptualisation: MBI, JR, GLC, VD. Development of the method, implementation: MBI, JR, GLC. Analysis of the results: MBI, JR, GLC, VD. Writing – original draft: MBI. Writing – reviewing and editing: MBI, JR, GLC, VD.

**Competing interests**

The authors declare that they have no conflict of interest.

**Supplement**

The supplement related to this work can be found online in the Supporting Information section at the end of this article.

**Acronyms**

BN – Bayesian network

DAG – Directed acyclic graph

CPT – Conditional probability table

CL – Confidence level

HP – Habitability pillar

RC – Risk criteria

RTH – Risk to habitability



*Acknowledgments.* This study was supported by the FUTURISKS project (Past-to-FUTURe coastal RISKS in Tropical French Overseas Island Territories: from impacts to solutions) funded by the French State under France 2030 under the reference ANR-22-POCE-0002. We thank Marissa Yates and Stéphane Costa for their helpful comments.

We acknowledge the assistance of AI language model ChatGPT in refining grammar of this manuscript.

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
