# Peer review of "Assessing atoll island future habitability in the context of climate"

_EGUsphere, 2024_

## Referee Comment (RC1)

**egusphere-2024-3884 | Journal relation: NHESS**    Submitted on 10 Dec 2024
**Assessing atoll island future habitability in the context of climate change using Bayesian networks**

Mirna Badillo-Interiano, Jérémy Rohmer, Gonéri Le Cozannet, and Virginie Duvat

DCISION: ACCEPT AS IT IS.

---

## Author Comment (AC2)

**Replies to Reviewers' comments on "Assessing atoll island future habitability in the context of climate change using Bayesian networks"**

We would like to thank the reviewers for their time and effort in providing detailed and constructive feedback. We have carefully considered each comment and revised the manuscript accordingly. Our responses are provided in this document.

**Reviewer #2**

*Summary of manuscript:*

*This study evaluates whether Bayesian Networks (BNs) based on expert judgments can effectively assess the integrated risk of climate change, and demonstrates a specific application to the habitability of atoll islands. A key methodological contribution is the translation of scores and confidence levels from Duvat et al. (2021) into probabilistic values, alongside the development of a Bayesian Network model. The manuscript presents a well-structured and clearly written methodology, with precise descriptions and strong visualizations. Strengths and limitations are discussed comprehensively. The study's scope aligns well with the focus of this special issue.*

We thank Reviewer #2 for the encouraging feedback and positive remarks.

**Minor comments:**

*-Lines 8-9: Reduce the use of "and" for improved readability.*

Thank you for this comment. This is now corrected.

*-**Line 11**: Specify the model type used by Duvat et al. to clarify whether it is a Bayesian Network.*

Thank you for this comment. We have revised the manuscript to clarify that the model type used by Duvat et al. (2021) is a multi-criteria expert-based assessment.

*-Lines 12, 51, 69 etc.: Remove brackets from citations of Duvat et al. -Line 24: AR6 citation is missing.*

Thank you for pointing this out. We have corrected the citation formatting and added the missing AR6 citation.

*-Lines 35ff: Land loss is not the only factor impacting food and water supply or economic activities— suggestion to include other relevant processes such as groundwater salinization.*

Thank you for this suggestion. We agree that multiple processes contribute to food and water insecurity and the disruption of economic activities. In response, we have revised the text to include other relevant processes such as soil and groundwater salinization, changes in rainfall patterns, and increased temperatures. We modified these sentences as follows:

[…Loss of coastal ecosystems and associated services resulting from the combination of global climate change and local anthropogenic disturbances will likely increase land loss, negatively impacting food and water supply as well as economic activities (Pratchett et al., 2008). In addition, other processes such as soil and groundwater salinization, changes in rainfall patterns, and increased temperatures are also expected to compromise resource availability and disrupt economic activities (Mycoo et al., 2022). This could challenge the ability of populations and ecosystems to recover and adapt in atoll settings.]

*-Lines 45ff: Clarify what is meant by "each driver of risk" and how it connects to Step 1 of the methodology.*

Thank you for the comment. We have clarified the meaning of "each driver of risk" by specifying that these drivers, such as sea level rise, population pressure, or extreme events, are the factors that can negatively affect the components defined in step one. We modified these sentences as follows:

[…Recognizing this complexity, many previous integrated assessments assessed present-day or future climate risk following a three-step approach. In a first step, a conceptual model identifying the different components of the studied system and their interlinkages was developed. Then, knowledge was collected using a multicriteria analysis to characterize the severity and confidence level of each risk factor. These factors, such as sea level rise, population pressure, or extreme events, are the drivers that can negatively affect the components defined in the first step. In a final step, this knowledge was aggregated. In the area of coastal risk assessments, the "Coastal Vulnerability Index" of (Gornitz et al., 1991) is a foundational example of such approaches.]

*-Table 1 Caption: The second and third sentences of the caption seem misplaced — consider integrating them into the main text or table itself. I would change "consideration of climate change" to "affected by climate change."*

Thank you for this suggestion. The second and third sentences of the caption are now integrated into the main text. Regarding the suggested phrasing, we propose using "Climate change-related variables" as we believe it appropriately reflects the nature of the variables considered in the model.

*-Figure 2 (and others): Improve resolution.*

This is now corrected.

*-Line 118: What is the meaning of "and stores the prior probability."?*

We agree that the phrasing was unclear. The prior probability is the marginal probability. When a node has no parents, it is associated with a probability table containing a marginal probability distribution, since it is not conditioned on any other variables. When a node has parents, it is associated with a conditional probability table, which contains a conditional probability distribution that depends on the states of its parent nodes. We have revised the sentence to clarify this as follows:

[…The structure is defined by nodes and arcs, representing the variables and the relationships between these variables, respectively (Figure 3). The arrow indicates the influence direction

from the *parent* node to the *child* node.  Each node in the network is associated with a probability table. When a node has no parents, it is associated with a probability table containing a marginal probability distribution, since it is not conditioned on any other variables. When a node has parents, it is associated with a conditional probability table, which contains a conditional probability distribution that depends on the states of its parent nodes.]

*-Figure 3: Specify the unit—does it represent meters of erosion or a 1–5 scale from Duvat et al.?*

Thank you for this comment. We have revised Figure 3 to clarify that the unit represents the 1-5 risk scale used by Duvat et al. (2021).

*-Equation 1: Define p(x_i) and I suggest to note in the text that conditional probabilities are used.*

Thank you for this suggestion. We have revised the text to define p(x_i) as the marginal probability of the variable x_i. We have also clarified that conditional probabilities are used. We modified the sentences as follows:

[…We focus on discrete BN, in which the product of the local probability distributions of each node results in the joint probability distribution function of all the variables X = $\{X_1, ... X_n\}$ in the graph Eq. (1):

$$P(X_1, ... X_n) = \prod_{i=1}^{n} p(X_i | pa(X_i)),$$
(1)

 p($X_i$) is the marginal probability of the variable $x_i$ , and pa($X_i$) are the parents of the variable $X_i$ (Pearl, 1988). The conditional probability distribution of the variable $X_i$ given its parents nodes is denoted by $p(x_i | pa(X_i))$. The joint probability distribution function links all the variables in the BN, therefore, any change can be propagated through the network.]

*-Line 145: "Malé" instead of "Male'."*

This is now corrected.

*-Table 2: Consider adding flooding from precipitation as a hazard in Malé, given high urbanization, coastal engineering, and inadequate drainage systems. Also, one could mention freshwater salinization as a key hazard in the Maldives.*

Thank you for this suggestion. We agree that pluvial flooding and freshwater salinization are important hazards in the Maldives. We have updated Table 2 to include them.

*-Figure 6: The x-axis label ("risk level") is the same as the plot titles—clarify the distinction. Replace "CL" with "Confidence Level" for clarity. Instead of using two dotted lines to indicate intervals, consider a transparent shaded area for better visualization.*

Thank you for this suggestion. We have clarified the distinction by changing the plot titles to "probability intervals". We have also modified the plot by adding "Confidence level" and the shaded area, as suggested.

*-Lines 224ff: Clearly state that the model is populated with these input data, rather than just used for validation. If validation means that aggregated risk to island habitability aligns with Duvat et al., be explicit about this definition to avoid ambiguity.*

Thank you for pointing out this ambiguity. We have revised the paragraph to clarify that the BN model is populated with input data from Duvat et al. (2021), as detailed in section 2.4.2. We have also clarified our definition of validation: By validation, we mean that the most likely risk levels estimated by the BN model are consistent with the risk levels assessed by Duvat et al. (2021). We modified the sentences as follows:

[…We  populated the BN model using expert knowledge and the assessment database from Duvat et al. (2021). We verified that the model structure, discretization of variables, and parametrization were consistent with expert knowledge and existing literature on atoll socio-ecosystems. To validate the model behaviour, we  compared  the most likely risk levels estimated by both the sub-networks and the entire network  with the risk levels assessed by the experts. This validation step confirmed that the BN model correctly reflects the expert-based evaluations. The validation process could be improved by integrating quantitative data such as measurements or numerical modeling results.]

*-Table 4: Ensure consistency in conditional probability notation (e.g., specify all dependencies or none, such as RCP vs. "Risk to island habitability = High/Very High").*

Thank you for pointing out this inconsistency. We have revised the table to ensure uniform conditional probability expressions. We modified the probabilistic queries as follows:

| Research question | Probabilistic query | Section |
|---|---|---|
| What is the probability of risk to habitability given a RCP scenario and a time horizon? |

P(Risk to island habitability \| RCP = 2.6/8.5 & Time horizon = 2050/2090) | 4.1 |
| What levels of risk could lead to adaptation limits? |

P(Risk to island habitability \| RCP = 2.6/8.5 & Time horizon = 2050/2090 & Risk criteria = 0/1/2/3/4/5) | 4.2 |
| Which risk factors are present when the risk to island habitability is high or very high? |  | 4.3 |

| | P(Risk criteria \| RCP = 2.6/8.5 & Time horizon = 2050/2090 & Risk to island habitability = High/Very high) | |
|---|---|---|
| To what extent is the risk to habitability reduced if we act on the risk factors that contribute the most? | In this experiment, we suppose the implementation of adaptation measures that reduce the risk factors to moderate levels. For example, Flooding <= 2:

 P(Risk to island habitability\| RCP = 2.6/8.5 & Time horizon = 2050/2090 & Flooding = 0, 1, and 2) | 4.4 |

*Major comments:*

*-Clearly define the literature gap. Line 40: Provide a clearer definition of integrated risk assessment. Is the literature gap identified in the caption of Table 1, i.e. no Bayesian Networks for small islands include integrated risk assessments? Or is it that integrated risk assessments for small islands are generally lacking? Does the statement in Line 67 contradicts this by suggesting that some BN studies with integrated climate risk assessments do exist — please clarify.*

Thank you for pointing out this ambiguity. The literature gap we highlight concerns the limited application of Bayesian Networks to conduct integrated risk assessments in small islands. Some studies have used BNs to assess hazards and adaptation strategies, however, to our best knowledge, no studies have applied BNs to perform integrated risk assessments that account for the combined effects of multiple climatic and non-climatic drivers. We have revised the manuscript to clarify this point as follows:

[…BNs explicitly integrate uncertainties, suggesting their potential to address climate change-related issues (Sperotto et al., 2017). Some applications include impact assessments of sea-level rise (Gutierrez et al., 2011; Yates and Le Cozannet, 2012) and coral reef degradation (Baldock et al., 2019), and evaluations of adaptation strategies (Hafezi et al., 2020; Phan et al., 2020).  Despite their growing application to climate change-related issues, only a few focused on integrated climate risk assessments (Catenacci and Giupponi, 2013). For small islands, applications remain limited, and to our best knowledge, integrated risk assessments are not available or have not yet been conducted.]

*-Terminology Precision (Lines 80ff): Ensure consistent and precise use of "risk" and "probability." It would be helpful to define key terms explicitly. For example: What constitutes risk? Are risk factors treated as variables, probabilities, or both? Does Research Question 1 focus on estimating the probability of island inhabitability? How is risk level determined? What are the drivers of risk? Later in the manuscript, risk criteria are introduced with a 1–5 scoring system. While the methodology clarifies the model and its variables, the Introduction would benefit from clearer definitions and framing to enhance coherence and reader understanding.*

Thank you for this insightful comment. We fully agree on the importance of precise and consistent use of terminology to avoid any confusion. In response, we have revised the Introduction to define the key terms. We have included the following definitions:

In our study, we followed the terminology used by Duvat et al. (2021) to ensure consistency.

**Risk**: Defined as the potential for adverse consequences to atoll socio-ecosystems from climate change, following the IPCC (2022) definition.

**Habitability pillars:** These are defined as the essential dimensions of habitability in small islands: available land, freshwater supply, food supply, settlements and infrastructure, and economic activities, following Duvat et al. (2021).

**Risk factors / Risk criteria**: here, we define risk factors or criteria as the main factors contributing to the risk to the habitability pillars. We used 14 risk factors identified by Duvat et al. (2021), including coastal erosion, flooding, fresh groundwater salinization, reduced reef fisheries production, loss of settlements, and reduction in tourism revenue. In our study, the risk factors are variables within our Bayesian network.

In the previous version, we used the term "Drivers of risk" to refer to "Risk factors". We have revised the manuscript to use only the term "Risk factors" to ensure consistency.

**Risk level**: This refers here to a scale describing the future additional risk from climate stressors (from "Undetectable" to "Very high").

**Probability of risk level**: This refers here to the probability that a given risk level will occur under specific conditions. The probability of risk level is calculated by the Bayesian Network and reflects the uncertainty about the future risk levels.

*-Section 4.4: Effectiveness of Adaptation Measures. The study does not apply concrete adaptation measures but rather evaluates risk reduction scenarios. Consider renaming Figure 10 to "Risk levels under different reduction scenarios" or "… under different risk reduction measures" (as already phrased elsewhere in the manuscript). Additionally, in Line 386, stating that the study "considers examples of adaptation measures" is misleading—reword to reflect that it assesses potential risk reductions that could be achieved through adaptation.*

Thank you for this pertinent comment. We agree that our study does not assess concrete adaptation or risk reduction measures, but rather explores risk reduction scenarios. While this is mentioned in the Discussion section, it was not reflected in Section 4.4, thank you for pointing this out. In response, we have revised the manuscript and figure to clarify that our analysis focuses on risk reduction scenarios. We modified the sentences as follows:

[… Section 4.4.  Risk reduction scenarios

In this experiment, the objective is to assess how the risk to habitability decreases when key risk factors are reduced, especially those previously identified as major contributors to the risk to habitability. To explore this, we assume  different risk reduction scenarios that could be achieved through adaptation, such as managed retreat and the implementation of measures to reduce flooding and coastal erosion. The impact of  such reductions is evaluated by calculating the probability of risk to habitability given a risk criteria level <= 2 under the RCP = 8.5 in 2090. In both islands, the results show that reducing flooding has a major impact on the risk to habitability (Figure 10). In Malé,  reducing flooding to moderate levels

could reduce the risk to habitability from high to moderate. These results suggest that an adaptation strategy focused on reducing flood risk only could already provide substantial benefits to preserve the habitability of Malé Island. Yet, focusing only on flood risk would not be sufficient in all contexts. For example, Figure 10 shows that  reducing multiple risks is necessary for Fogafale to keep the risk to habitability at a moderate level.]

Orléans,
June 6, 2025
M. Badillo-Interiano[1] on behalf of the co-authors

[1]BRGM, 3 av. C. Guillemin – 45060 Orléans Cedex 2 - France

---

## Referee Report (RR1)

The manuscript has improved significantly, and the authors have addressed all previous review comments thoroughly. I have additional minor comments, highlighted in yellow:

-Terminology Precision (Lines 80ff): Ensure consistent and precise use of "risk" and "probability." It would be helpful to define key terms explicitly. For example: What constitutes risk? Are risk factors treated as variables, probabilities, or both? Does Research Question 1 focus on estimating the probability of island inhabitability? How is risk level determined? What are the drivers of risk? Later in the manuscript, risk criteria are introduced with a 1–5 scoring system. While the methodology clarifies the model and its variables, the Introduction would benefit from clearer definitions and framing to enhance coherence and reader understanding.

Thank you for this insightful comment. We fully agree on the importance of precise and consistent use of terminology to avoid any confusion. In response, we have revised the Introduction to define the key terms. We have included the following definitions: In our study, we followed the terminology used by Duvat et al. (2021) to ensure consistency.

**Risk:** Defined as the potential for adverse consequences to atoll socio-ecosystems from climate change, following the IPCC (2022) definition. **Habitability pillars:** These are defined as the essential dimensions of habitability in small islands: available land, freshwater supply, food supply, settlements and infrastructure, and economic activities, following Duvat et al. (2021). **Risk factors / Risk criteria:** here, we define risk factors or criteria as the main factors contributing to the risk to the habitability pillars. We used 14 risk factors identified by Duvat et al. (2021), including coastal erosion, flooding, fresh groundwater salinization, reduced reef fisheries production, loss of settlements, and reduction in tourism revenue. In our study, the risk factors are variables within our Bayesian network. In the previous version, we used the term "Drivers of risk" to refer to "Risk factors". We have revised the manuscript to use only the term "Risk factors" to ensure consistency. **Risk level:** This refers here to a scale describing the future additional risk from climate stressors (from "Undetectable" to "Very high"). **Probability of risk level:** This refers here to the probability that a given risk level will occur under specific conditions. The probability of risk level is calculated by the Bayesian Network and reflects the uncertainty about the future risk.

-The term climate stressors is used here to define the risk level, but it is not mentioned anywhere else in the manuscript and not defined. Maybe risk factors could be used instead to be coherent?

-I cannot unambiguously find the case site Tabiteuea, North Tarawa, Kiribati. Maybe a more detailed description where this island is is necessary? I could only find Tabiteuea as an atoll island, or a church called Kiribati Uniting Church - Tabiteuea on Tabiteuea, North Tarawa on Google Earth, but the satellite image does not seem to match with the one from the paper. I would like to know the coordinates to identify the case site.

-Typo for one of the sites: Fongafale, Tuvalu. It is written as Fogafale in the manuscript.